# m^6^A Modified Short RNA Fragments Inhibit Cytoplasmic TLS/FUS Aggregation Induced by Hyperosmotic Stress

**DOI:** 10.3390/ijms222011014

**Published:** 2021-10-12

**Authors:** Ryoma Yoneda, Naomi Ueda, Riki Kurokawa

**Affiliations:** Division of Biomedical Sciences, Research Center for Genomic Medicine, Saitama Medical University, 1397-1 Yamane, Hidaka-shi, Saitama 350-1241, Japan; n_ueda@saitama-med.ac.jp

**Keywords:** TLS/FUS, RNA m^6^A modification, LLPS, lncRNA, pncRNA-D

## Abstract

Translocated in LipoSarcoma/Fused in Sarcoma (TLS/FUS) is a nuclear RNA binding protein whose mutations cause amyotrophic lateral sclerosis. TLS/FUS undergoes LLPS and forms membraneless particles with other proteins and nucleic acids. Interaction with RNA alters conformation of TLS/FUS, which affects binding with proteins, but the effect of m^6^A RNA modification on the TLS/FUS–RNA interaction remains elusive. Here, we investigated the binding specificity of TLS/FUS to m^6^A RNA fragments by RNA pull down assay, and elucidated that both wild type and ALS-related TLS/FUS mutants strongly bound to m^6^A modified RNAs. TLS/FUS formed cytoplasmic foci by treating hyperosmotic stress, but the cells transfected with m^6^A-modified RNAs had a smaller number of foci. Moreover, m^6^A-modified RNA transfection resulted in the cells obtaining higher resistance to the stress. In summary, we propose TLS/FUS as a novel candidate of m^6^A recognition protein, and m^6^A-modified RNA fragments diffuse cytoplasmic TLS/FUS foci and thereby enhance cell viability.

## 1. Introduction

Translocated in LipoSarcoma/Fused in Sarcoma (TLS/FUS) is a nuclear RNA/DNA binding protein whose mutations lead to amyotrophic lateral sclerosis (ALS). In particular, mutations in the C terminal nuclear localization signal (NLS) are related to severe ALS onset and symptoms [1,2,3]. Mutations in NLS result in mislocalization of TLS/FUS in the cytoplasm and force TLS/FUS to form cytotoxic aggregates via liquid-liquid phase separation (LLPS) [4].

LLPS has received much attention in the field of RNA binding proteins (RBPs). Many RBPs have an intrinsically disordered region (IDR), the region which does not fold into high-order structures [5,6,7,8]. IDRs interact by themselves, forming membraneless particles, which could isolate contents from external environments. Subcellular organelles, such as nucleoli, stress granules, P bodies, and paraspeckles, are formed as a result of LLPS [9]. LLPS is involved in various cellular functions, such as stress response, assembly of ribosomes, isolation of RNAs and proteins, and axonal transport [10,11].

LLPS is deeply related to diseases, in particular, neurodegenerative diseases including Alzheimer’s, Parkinson’s, Huntington diseases, and ALS [12]. The aggregates of RBPs formed through LLPS provoke neuronal cell apoptosis or necrosis, and subsequently disorders in neuronal systems [13,14]. Fibrilization of Tau protein, which are abundantly expressed in neurons, are often seen in the patients of Alzheimer’s diseases [15,16], and aggregates of TDP-43 and TLS/FUS are predicted to be the cause of ALS [3,17,18].

LLPS of TLS/FUS are regulated by various factors including post-translational modification, external environment, or interaction with proteins and RNAs. The arginine methylation of TLS/FUS alters cation–pie interaction [19], and phosphorylation of TLS/FUS IDR disrupts aggregation [20]. The external environment, such as salt concentration, pH, temperature, or shear stress, could affect TLS/FUS LLPS [19,21]. Interaction with karyopherin β2 or Hsp27 disrupts the LLPS of TLS/FUS [22,23]. In addition, binding with RNA could either promote or inhibit the LLPS of TLS/FUS. Generally, interaction with RNA inhibits TLS/FUS LLPS, but particular RNAs, such as *NEAT1*, could promote LLPS [24]. Not all the LLPS results in cytotoxicity, but properly arranged LLPS of TLS/FUS is necessary for DNA repair initiation [25].

m^6^A modification is one of the most abundant RNA modifications found in mRNAs as well as in noncoding RNAs. Its function covers RNA degradation, translational inhibition, and alteration of interaction with proteins [26]. This modification is recognized by the protein family with a YTH domain, and the well-known reader protein YTHDF2 undergoes LLPS by interacting with m^6^A-modified mRNA [27].

We previously reported that TLS/FUS is recruited to the promoter region of cyclin D1 by long noncoding RNA (lncRNA) named *pncRNA-D*, and interaction with *pncRNA-D* dynamically changes the conformation of TLS/FUS [28,29,30]. We and others demonstrated that TLS/FUS binds to short RNAs in a length- and sequence-specific manner [28,29]. Collectively, we hypothesized that interaction with RNAs and subsequent conformational alteration would affect LLPS of TLS/FUS.

In this study, we validated the binding specificity of TLS/FUS to short RNA fragments derived from *pncRNA-D*, and evaluated the effect of RNA m^6^A modification. Intriguingly, both wild type and ALS-related TLS/FUS mutants demonstrated stronger binding to m^6^A-modified RNA fragments. This interaction decreased cytoplasmic TLS/FUS foci in cultured cell lines, and subsequently enhanced cell viability. In conclusion, we propose TLS/FUS as a novel m^6^A reader candidate, and m^6^A-modified short RNA fragments enhance the cell viability by diffusing cytoplasmic TLS/FUS aggregates formed by LLPS.

## 2. Results

### 2.1. TLS/FUS Binds Intensely to m^6^A-Modified RNA Fragments

We first examined if RNA m^6^A modification could alter the interaction between RNA and TLS/FUS. For this purpose, we generated two RNA fragments derived from *pncRNA-D*, an lncRNA expressed from cyclin D1 promoter (Figure 1A, black box, [29]). In a previous study, we defined GGACU motif at the 467 nt of *pncRNA-D* is m^6^A modified, and 20 nt RNA fragment (named Fragment 6, Figure 1A) around this motif robustly binds to TLS/FUS [30]. This time, we selected an additional GGACC motif at the position of 249 nt of *pncRNA-D*, which is speculated to be m^6^A modified (Fragment 3, Figure 1A).

Biotinylated RNA fragments with or without m^6^A modification were incubated with *E. coli* overexpressed strep-GFP-TLS/FUS (either wild type (WT) or ALS-related mutants, purified with strep-tag). As a result of RNA pull down assay, WT TLS/FUS showed the highest binding specificity. The strongest interaction was observed with m^6^A-modified Fragment 6, and weak binding to unmodified Fragment 6 (Figure 1B, WT). WT TLS/FUS did not bind to Fragment 3, regardless of m^6^A modification. On the other hand, ALS-related TLS/FUS mutants demonstrated lower binding specificity compared to WT. The R521G mutant bound intensely to Fragment 6 but also to Fragment 3 to some extent, and R495X mutant, which completely lacks NLS, had a similar affinity to both Fragment 3 and 6 (Figure 1B, R495X and R521G). Of note, TLS/FUS mutants also exhibited stronger binding to m^6^A-modified fragments (Figure 1C). These data suggest the possibility of TLS/FUS as a novel m^6^A reader protein, and that NLS plays an important role in RNA recognition.

### 2.2. m^6^A-Modified RNA Fragments Did Not Promote LLPS of TLS/FUS

Next, RNA fragments were transfected to HAP1 cells in order to validate the effect of m^6^A-modified RNAs on LLPS of TLS/FUS in the cell. Cy5-labeled Fragment 6 (with or without m^6^A, Figure 1A) was transfected to HAP1 cells. Cy5 signals indicated that RNA fragments were transfected to most of the cells (Figure 2A, red signals), which reflects the high transfection efficiency of the experiment. Transfected RNAs formed particles mainly in the cytoplasm and some in the nucleus, but WT TLS/FUS still localized to the nucleus, and did not colocalize with RNAs in the cytoplasm (Figure 2A).

We then conducted the same experiment with TLS/FUS mutants-expressing cell lines. As a first step, we constructed stable cell lines that express GFP-tagged TLS/FUS mutants (R495X or R521G). Overexpression plasmids were transfected to TLS/FUS knockout HAP1 cells, since an excess amount of TLS/FUS triggers cytotoxicity [3,31]. GFP signals were confirmed by both fluorescent microscopy and Western blot analysis (Figure 2B,C, Appendix A Appendix A). R495X mutants were diffused and formed a small number of particles in the cytoplasm (Figure 2B, green signals), and it was difficult to detect colocalization with RNA fragments (Figure 2B, white arrowhead). Most of the R521G mutants were localized to the nucleus like WT but partially formed particles in the cytoplasm (Figure 2C, green signals), and colocalized with m^6^A Fragment 6 (Figure 2C, white arrowheads). However, only 0.95 ± 0.14% of the cells had R521G particles colocalized with RNA, and the majority of the particles localized independently of RNA. These results suggest that RNA Fragment 6 did not promote LLPS of TLS/FUS, regardless of m^6^A modification.

### 2.3. Cytoplasmic TLS/FUS Aggregation Is Inhibited by m^6^A-Modified RNA Transfection

Since the 13 nt RNA fragment named *pncRNA-D* 5′1-1 inhibited the aggregation of TLS/FUS in vitro [21,28], we expected that m^6^A-modified RNA fragments might also function in the reduction of LLPS particles. In order to induce TLS/FUS LLPS in the cytoplasm, HAP1 cells were treated with 0.4 M sorbitol, since hyperosmotic stress caused by sorbitol treatment induces TLS/FUS foci formation [32]. As a previous study, cytoplasmic TLS/FUS foci were observed in most of the cells treated with 0.4M sorbitol (Figure 3A, bottom panels). Then, we transfected RNA fragments prior to sorbitol treatment (Figure 3B), and evaluated the effect on cytoplasmic TLS/FUS aggregation. As a result, transfection of RNA fragments decreased cells with cytoplasmic TLS/FUS foci, except for unmodified Fragment 3 (Figure 3C,D). Remarkably, cells transfected with m^6^A-modified Fragment 6 had the highest reduction efficiency (Figure 3D, pink box, Appendix A Appendix A), and the data clearly correlated with the binding intensity as in Figure 1. In addition, the number of TLS/FUS foci per cell was also decreased by m^6^A RNA transfection (Figure 3E, green and pink plots). These results indicate the inhibitory effect of m^6^A-modified RNA fragments on cytoplasmic LLPS of TLS/FUS. Since TLS/FUS localizes to stress granules (SGs) with cellular stresses [32], the formation of SGs and TLS/FUS colocalization was observed. As a result, the transfection of RNA fragments slightly decreased the formation of SGs (Appendix A Appendix A), and led to increased TLS/FUS foci localized independently of SGs (Appendix A Appendix A, black boxes).

Similarly, the effect of m^6^A-modified RNA fragments on cytoplasmic aggregation of TLS/FUS mutants was observed. Because R521G and R495X mutants scarcely formed foci in the cytoplasm (Figure 2B,C, [33]), the two mutants were also treated with 0.4M sorbitol to induce cytoplasmic foci. The GFP-R521G-expressing stable cell line did not form sufficient cytoplasmic foci for analyses even after sorbitol treatment (Appendix A Appendix A); therefore, the data of the R495X mutant is described. GFP-R495X-expressing stable cells formed cytoplasmic foci after 0.4M sorbitol treatment (Figure 4A, top panels). The percentage of cells with cytoplasmic foci was reduced by transfection of any kind of the RNA fragments examined (Figure 4A,B, Appendix A Appendix A). This was in good agreement with the low binding specificity observed in Figure 1, where R495X mutant bound to all four RNA fragments. The number of the cytoplasmic foci per cell was also decreased by transfecting RNA fragments, and the cells transfected with m^6^A-modified fragments had smaller numbers of cytoplasmic foci compared to unmodified fragments (Figure 4C, green and purple plots). The results indicated that m^6^A-modified RNAs inhibited cytoplasmic LLPS of the TLS/FUS R495X mutant as well as WT.

### 2.4. m^6^A-Modified RNA Fragment Has a Different Effect on Localization of TLS/FUS-Interacting Proteins

Mutated TLS/FUS often colocalize with other ALS-related proteins in the cytoplasm [34,35], and this time, we sought for novel proteins that interact with TLS/FUS. As a result of TLS/FUS coimmunoprecipitation and mass spectrometry, we found that Matrin3 and ZAP3 bound significantly to TLS/FUS (Appendix A Appendix A). Both proteins are usually localized to the nucleus (Figure 5A and Figure 6A, top panels), but 0.4M sorbitol treatment led to mislocalization of Matrin3 and ZAP3 in the cytoplasm, and colocalized with TLS/FUS. Cytoplasmic colocalization of Matrin3 and mutated TLS/FUS has already been reported [36], but this is the first report that indicated interaction between TLS/FUS and ZAP3. We then confirmed the effects of RNA fragments on Matrin3 or ZAP3 foci formation and colocalization with TLS/FUS. Most of the Matrin3 foci were colocalized with TLS/FUS in the sorbitol-treated HAP1 cells (Figure 5C), but the transfection of unmodified Fragment 6 increased the proportion of Matrin3 foci without TLS/FUS (Figure 5C, white box). Transfection of m^6^A-modified Fragment 6 reduced the number of cytoplasmic Matrin3 foci (Figure 5B), but we still detected colocalization with remaining cytoplasmic TLS/FUS foci (Figure 5A, white arrowheads, Figure 5C). Meanwhile, RNA fragments had little effect on cytoplasmic ZAP3 aggregation (Figure 6B), and ZAP3 tended to form foci independently of TLS/FUS compared to Matrin3 (Figure 6A,C).

We also examined the effect of RNA fragments on well-known TLS/FUS interactome TDP-43. HAP1 cells treated with sorbitol demonstrated almost all the cytoplasmic TDP-43 foci localized with TLS/FUS (Appendix A Appendix A). Transfection of unmodified Fragment 6 resulted in an increased proportion of TLS/FUS foci without TDP-43, and cells transfected with m^6^A-modified Fragment 6 had a smaller number of TDP-43 foci compared to ‘No RNA’ control (Appendix A Appendix A).

### 2.5. m^6^A-Modified RNA Fragments Enhance the Viability of Cells Treated with Sorbitol

Finally, the effect of m^6^A-modified RNA fragments on cell viability was investigated. We transfected HAP1 cells and the GFP-R495X stable cell line with four RNA fragments as in Figure 3, and treated them with 0.4M sorbitol. After an hour of incubation, the media was exchanged with a fresh media without sorbitol for cell recovery, and the cell viability was measured at each time point indicated in Figure 7A. While transfection of RNA fragments did not alter the viability in the cells without treatment (Figure 7B), it had remarkable effects on the cells treated with 0.4M sorbitol. In sorbitol-treated HAP1 cells, transfection of Fragment 3 with or without m^6^A modification exhibited slightly higher but not significant cell viability compared to control cells (Figure 7C, WT, blue lines). On the other hand, cells transfected with Fragment 6 showed significantly enhanced cell viability (Figure 7C, WT, red lines). Notably, m^6^A-modified Fragment 6 demonstrated higher viability than unmodified Fragment 6 (Figure 7C, WT, red broken line).

In addition, the cell viability of the sorbitol-treated GFP-R495X stable cell line was potentiated by transfection of m^6^A-modified Fragments 3 and 6 (Figure 7C, R495X, blue and red broken lines). Taken together, m^6^A-modified RNA fragments were able to promote recovery from cellular stress both in WT HAP1 cells and the GFP-R495X stable cell line.

## 3. Discussion

Here, we report that TLS/FUS preferentially bound to the m^6^A-modified RNA fragments, and mutations in NLS reduced the RNA binding specificity of TLS/FUS. TLS/FUS formed cytoplasmic foci by treating hyperosmotic stress, and these foci were diminished by transfection of RNA fragments, especially m^6^A-modified fragments. Moreover, the cells transfected with m^6^A RNA fragments had higher resistance to hyperosmotic stress. In summary, TLS/FUS is proposed to be a novel m^6^A reader candidate, and m^6^A-modified RNA fragments enhance cell viability by interfering in cytoplasmic TLS/FUS aggregation.

NLS of TLS/FUS is an essential domain for its nuclear localization, and multiple NLS mutations are found in both familial and sporadic ALS patients [18]. It is well known that mutations in NLS alter binding with proteins [22,37], but its effect on RNA binding remains to be elucidated. Hoell et al. reported that the TLS/FUS R521G mutant had a different RNA binding profile compared to WT in HEK293 cells [38], but they also claim that this was probably be due to mislocalization of the TLS/FUS mutant to the cytoplasm. In addition, a recent study revealed that TLS/FUS changes its interactome depending on its state, diffused or aggregated via LLPS [39]. Even in our in vitro experiment, R495X mutant exhibited the lowest specificity to the RNAs examined in this study, which implies NLS of TLS/FUS determines the specificity of RNA binding. Further studies are required to determine if NLS could alter binding with longer RNAs, but we conclude that TLS/FUS NLS take part in recognition of RNA.

We have to emphasize that although TLS/FUS does not contain a YTH domain, which is regarded as an essential domain for m^6^A recognition [40,41], either WT or mutated TLS/FUS strongly bound to m^6^A-modified RNA fragments. In the previous paper, we inferred that TLS/FUS could be an m^6^A reader protein [30], and to the best of our knowledge, this is the first report describing the TLS/FUS binding specificity to m^6^A-modified RNAs. Even in the R495X mutants, which had the lowest RNA binding specificity, it preferentially bound to m^6^A-modified RNA fragments. Many of the well-known m^6^A reader proteins localize in the cytoplasm (YTHDF1-3 and YTHDC2), and only YTHDC1 functions in the nucleus [42]. Collectively, we insist on TLS/FUS as a novel candidate of m^6^A reader protein localized in the nucleus, and further experiments will be conducted to determine which domain of TLS/FUS is responsible for recognition of the m^6^A modification.

The relationship between m^6^A modification and LLPS is largely unknown. The LLPS of the m^6^A reader protein family, YTHDF proteins, is promoted by m^6^A-modified mRNAs [27,43], and knockdown of the m^6^A modification protein METTL14 reduces YTHDF2 colocalized in stress granules [44]. In this study, we obtained a different effect, in that m^6^A-modified RNA fragments bound to TLS/FUS and this interaction inhibited LLPS. The LLPS of TLS/FUS can be either promoted or inhibited by the addition of RNAs, depending on their sequences and secondary structures [24,28,38]. *pncRNA-D* is the lncRNA of our interest, and the four RNA fragments used in this study were derived from *pncRNA-D*. *pncRNA-D* binds to C terminal RGG2-Zinc Finger-RGG3-NLS domains of TLS/FUS [29], and this interaction changes the conformation of TLS/FUS [45]. The interaction between m^6^A-modified RNA fragments and TLS/FUS is also expected to change the protein structure, which could subsequently disperse TLS/FUS foci formed by LLPS.

RNA fragments had distinct effects on Matrin3, ZAP3, or TDP-43 cytoplasmic aggregation. Matrin3 and TDP-43 are nuclear RNA binding proteins (RBPs) and their mutations are found in both sporadic and familial ALS [46,47,48,49,50]. Cytoplasmic Matrin3 and TDP-43 foci were reduced by RNA fragments’ transfection. This was conceivably the result of diffused cytoplasmic TLS/FUS foci, because TLS/FUS foci could sequester other ALS-related nuclear RBPs in the cytoplasm [34,35]. Meanwhile, ZAP3 is a large nuclear protein with over 2000 amino acids, which interacts with serine/threonine protein phosphatase PP1 [51]. Computational analysis revealed that it had several predicted IDRs with a length of around 200 amino acids (Appendix A Appendix A). Accordingly, ZAP3 is able to undergo LLPS and form cytoplasmic foci independently of TLS/FUS, presumably due to its weak interaction with TLS/FUS (Appendix A Appendix A).

Cytoplasmic aggregates of RBPs are often observed in neurons of neurodegenerative disease patients, and are speculated to be the cause of the diseases [17,52,53]. Dissolving these aggregates is anticipated to lead to the treatment of the diseases, and 1,6-hexanediol is one of the most widely used reagents in order to disperse LLPS of RBPs in vitro [54,55]. Nevertheless, 1,6-hexanediol can provoke defects in cellular functions, such as impairment of kinase and phosphatase activities, and immobilization of chromatins [56,57]. We also observed that HAP1 cells cannot survive in 4% 1,6-hexanediol for 5 min or longer (data not shown). Therefore, novel reagents or drugs that could disperse cytoplasmic RBP aggregates are strongly desired.

Recently, lncRNAs have become the target sequence of oligonucleotide therapeutics [58,59]. For example, lncRNA PRAL (p53 Regulation-Associated LncRNA) interacts with Hsp90 and decreases tumor growth [60]. Oligonucleotide therapeutics targeting neurodegenerative diseases are now under development, but the cytotoxicity and off target effects are major concerns [61]. The 20 nt m^6^A-modified RNA fragments used in this study were able to potentiate cell viability after hyperosmotic stress without causing detectable cellular defects in normal cells. In conclusion, we propose the possibility of the m^6^A-modified RNA fragments being the seed sequences for oligonucleotide therapeutics targeting cytoplasmic TLS/FUS aggregation.

## 4. Materials and Methods

### 4.1. Cell Culture and Stable Cell Preparation

HAP1 cell culture was conducted as previously described [30]. TLS/FUS mutant (R495X and R521G) coding sequences were cloned into pAcGFP1-C3 vector (632482, Clontech Laboratories, CA, USA) with primers (R495X sense and antisense primers, 5′-ACC GCT CGA GAT GGC CTC AAA CGA TTA TAC-3′, 5′-TAT ACT GCA GTC AGA AGC CTC CAC G-3′; R521G sense and antisense primers, 5′-ACC GCT CGA GAT GGC CTC AAA CGA TTA TAC-3′, 5′-TAT ACT GCA GCT AAT ACG GCC TCT CCC TGC C-3′), and the plasmids were transfected to TLS/FUS-KO HAP1 cells by Lipofectamine 3000 (L3000015, Invitrogen, Carlsbad, CA, USA) according to the manufacturer’s protocol. After 48 h of incubation, GFP signals were confirmed by fluorescent microscopy, and the cells were treated with IMDM with 1mg/mL G418 (09380-86, Nacalai Tesque, Kyoto, Japan). Medium was changed every two days, and cultured for 10 days. Then, the GFP-positive and propidium iodide-negative cells were isolated to single cells in a 96-well plate by FACS (SH800Z, SONY). After one week, cells with strong GFP signals were chosen and the genome sequence was examined to confirm that they had the GFP-TLS/FUS mutated sequence in the genome.

### 4.2. RNA Pull Down Assay and Overexpression of GFP-TLS/FUS

The RNA pull down assay was performed as previously described [29] with slight modifications. Biotinylated 20 nt RNA fragments (Fragment 3 (with or without m^6^A modification), and Fragment 6 (with or without m^6^A modification)) were generated and purchased from Hokkaido System Science (Sapporo, Japan). RNA fragments were incubated with purified GFP-TLS/FUS. For GFP-tagged TLS/FUS overexpression, pAcGFP1-C3 vector with coding sequences of WT, R495X, or R521G was amplified with primers (WT and R521G sense and antisense primers, 5′-ATG GAT CCA GTG AGC AAG GGC GCC-3′, 5′-TAT ACT GCA GCT AAT ACG GCC TCT CCC TGC-3′; R495X sense and antisense primers, 5′-ATG GAT CCA GTG AGC AAG GGC GCC-3′, 5′-TAT ACT GCA GTC AGA AGC CTC CAC G-3′) and inserted into the pASK-IBA5plus vector (IBA, Göttingen, Germany). Overexpression and purification was performed as previously described [62].

### 4.3. Western Blot Analysis

Western blot analysis was performed as previously described [62]. Antibodies were anti-ACTB mAb (017-24551, Wako Pure Chemical, Osaka, Japan), anti-GFP (GF-200, Nacalai Tesque), and anti-TLS/FUS (ab154141, Abcam, MA, USA) for primary antibodies at the concentration of 1:2000, anti-rabbit IgG HRP-linked antibody (7074S, Cell Signaling Technology, Danvers, MA, USA) and anti-mouse immunoglobulins/HRP (P0161, DAKO, Glostrup, Denmark) for secondary antibodies at the concentration of 1:5000.

### 4.4. Immunocytochemistry (ICC) Assay

First, 5 × 10^3^ cells/chamber HAP1 cells were cultured in an 8-well slide and chamber (192-008, Watson, Tokyo, Japan). Cells were fixed with 4% PFA in PBS (26123-55, Nacalai Tesque), washed with PBS three times, and blocked with blocking buffer (5% skim milk with 0.3% triton X-100). Primary antibodies (anti-TLS/FUS (ab154141, Abcam, Waltham, MA, USA), anti-ZAP3 (A304-038A, Bethyl, Montgomery, TX, USA), anti-Matrin3 (A300-590A, Bethyl)) were diluted with blocking buffer at 1:1000, anti-TIA-1 (ab140595, Abcam) and anti-TDP-43 (12892-1-AP, Proteintech, Rosemont, IL, USA) were diluted at 1:250, and incubated at 4 °C for 16 h. Cells were washed for three times, and incubated with secondary antibodies (AlexaFlour 647 goat anti-rabbit IgG (A21244, Invitrogen) and Alexa Fluor 555 donkey anti-mouse IgG (A31570, Invitrogen), diluted with blocking buffer at 1:1000) at 37 °C for 1 h. Cells were washed again and mounted with Vectashield (H-1800, Vector Laboratories, Burlingame, CA, USA), and the cells were observed under fluorescent microscopy (BZ-X700).

### 4.5. RNA Induction and Sorbitol Treatment

HAP1 cells were transfected with 30 pmol RNA fragments with JetMESSENGER (150-001, Polyplus, Illkirch, France) according to the manufacturer’s protocol. After 4 h of incubation, the media was changed to fresh new media, and the cells were subsequently used for ICC or cell viability assay.

### 4.6. Cell Viability Assay

HAP1 cells were harvested in a 96-well plate at 5 × 10^3^ cells/well. The next day RNA fragments were transfected as above, and treated with or without sorbitol for 1 h. The cell viability was measured at each indicated time point in Figure 7A with Cell Count Reagent SF (07553-15, Nacalai Tesque) according to the manufacturer’s instruction.

### 4.7. Statistical Analysis

Data are presented as means ± standard deviations. Pairwise differences were identified using two-tailed Student *t* test and were considered significant when *p* < 0.05.

## Figures and Tables

**Figure 1 ijms-22-11014-f001:**
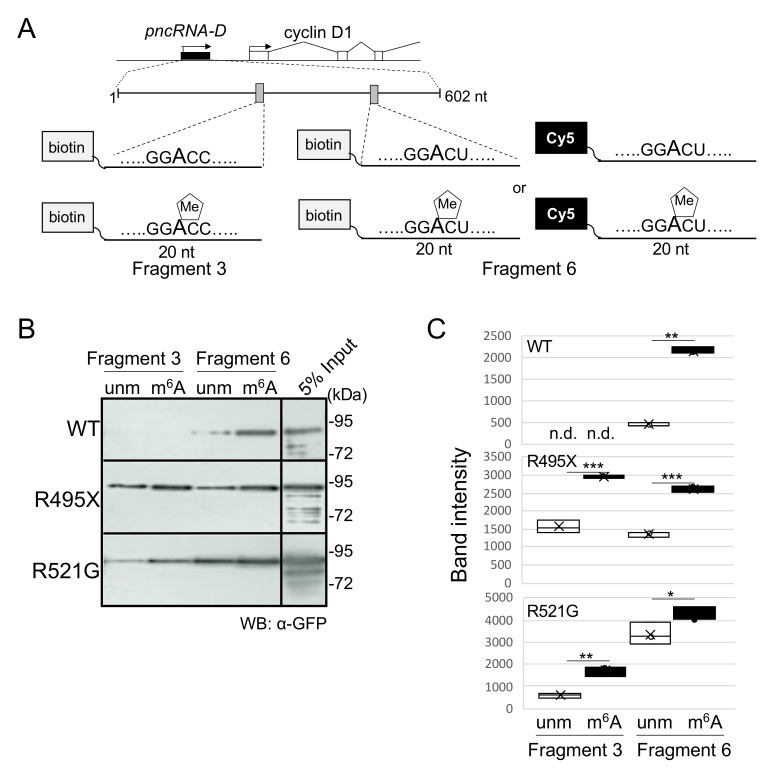
TLS/FUS bound preferentially to m^6^A-modified RNA fragments. (**A**) Schematic drawing of the region around the cyclin D1 promoter. Positions of RNA fragments used in this study are described. Two sequences (Fragments 3 and 6) were selected from the lncRNA *pncRNA-D* (black box). Biotinylated RNA sequences were prepared with or without m^6^A modification, and for Fragment 6, we also prepared sequences labeled with Cy5. (**B**,**C**) RNA pull down assay followed by Western blot analysis. Biotinylated RNA fragments were incubated with purified strep-GFP-TLS/FUS (WT or mutated) proteins, and the band intensities were quantified as in (**C**). n.d., not detected; unm, unmodified fragments; m^6^A, m^6^A modified fragments. *n* = 3. * *p* < 0.05, ** *p* < 0.01, *** *p* < 0.005.

**Figure 2 ijms-22-11014-f002:**
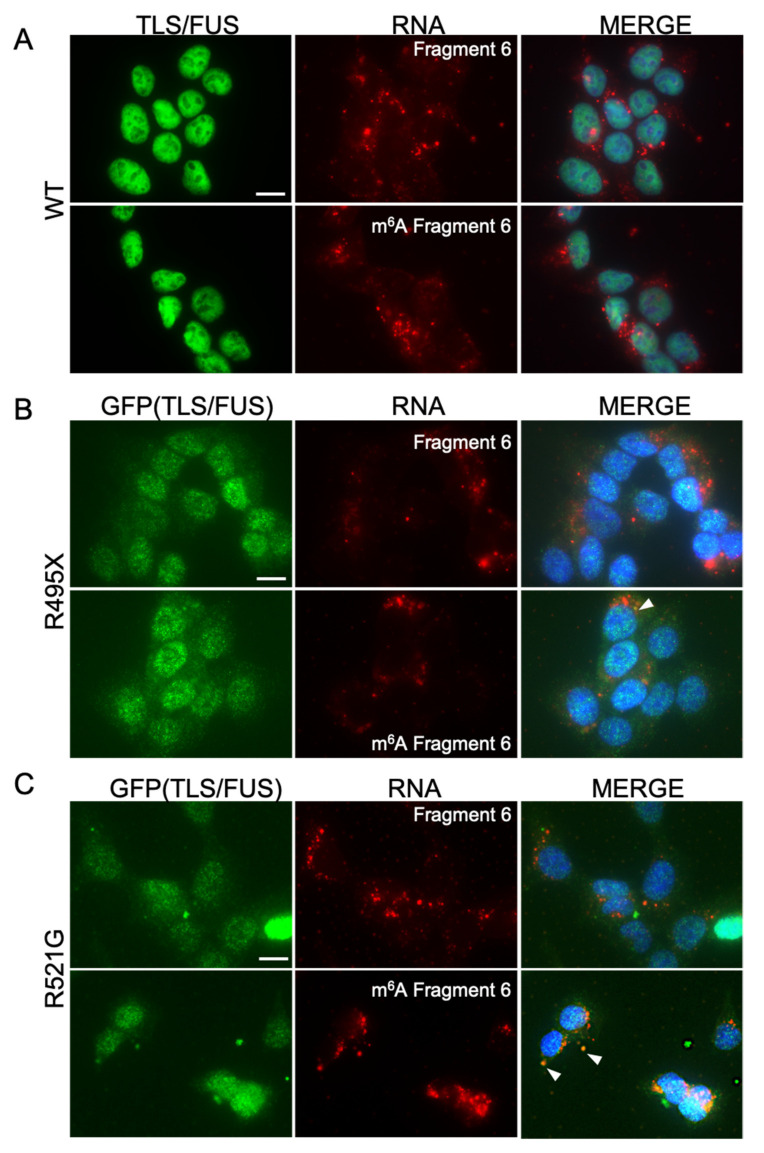
RNA fragments did not promote cytoplasmic TLS/FUS aggregation in HAP1 cells. (**A**) HAP1 cells were transfected with Cy5-labeled RNA Fragment 6 with or without m^6^A modification. Representative images of ICC are shown. MERGE images indicate the layered images of TLS/FUS, RNA, and DAPI (for nuclei staining). Scale bar = 10 μm. (**B**,**C**) The same experiment as in (**A**) was conducted with stable cell lines expressing GFP-R495X (**B**) or R521G (**C**). White arrowheads, colocalized foci of TLS/FUS and RNA. Scale bars = 10 μm.

**Figure 3 ijms-22-11014-f003:**
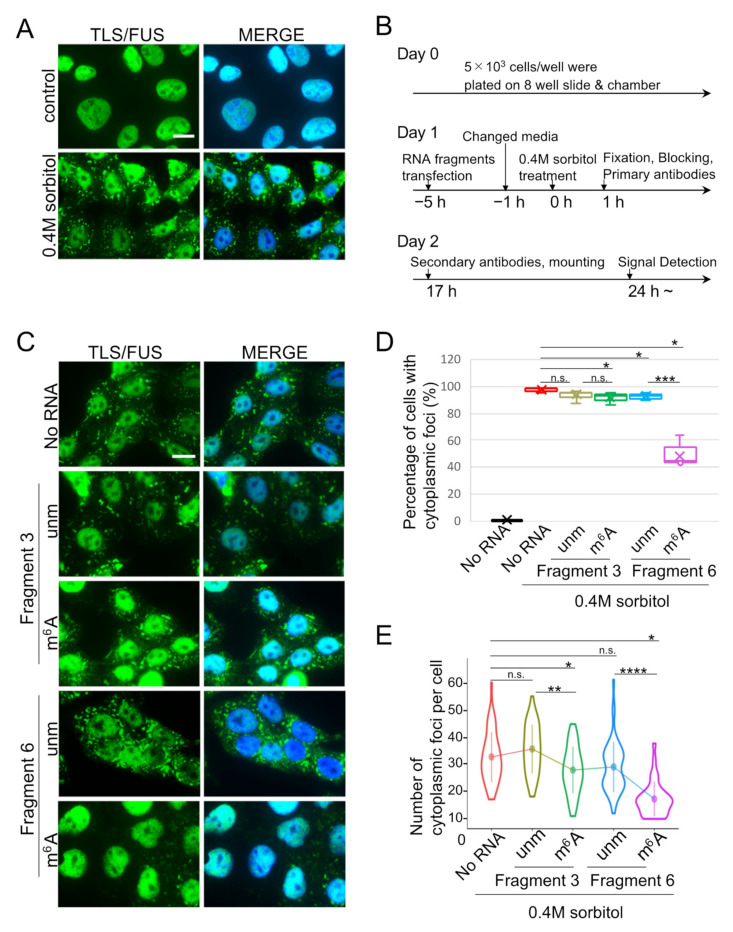
RNA fragments reduced cytoplasmic TLS/FUS foci after sorbitol treatment. (**A**) Representative ICC images of HAP1 cells with or without 0.4M sorbitol treatment for 1 h. Scale bar = 10 μm. (**B**) Experimental design of (**C**). RNA fragments were transfected prior to 0.4M sorbitol treatment, and ICC was conducted after 1 h of sorbitol treatment. (**C**) RNA Fragments 3 and 6 with or without m^6^A modification were transfected to HAP1 cells as in (**B**), and followed by sorbitol treatment and ICC. Representative images of ICC are shown. MERGE images indicate the layered images of TLS/FUS and DAPI (for nuclei staining). Scale bar = 10 μm. (**D**,**E**) Percentage of cells with cytoplasmic TLS/FUS foci (**D**) or number of cytoplasmic foci per cell (E) were counted and quantified using images in (**C**) and Appendix A Appendix A. *n* = 10 (**D**) or 50 (**E**) for each sample. n.s., not significant; unm, unmodified fragments; m^6^A, m^6^A modified fragments. * *p* < 0.05, ** *p* < 0.01, *** *p* < 0.005, **** *p* < 0.001.

**Figure 4 ijms-22-11014-f004:**
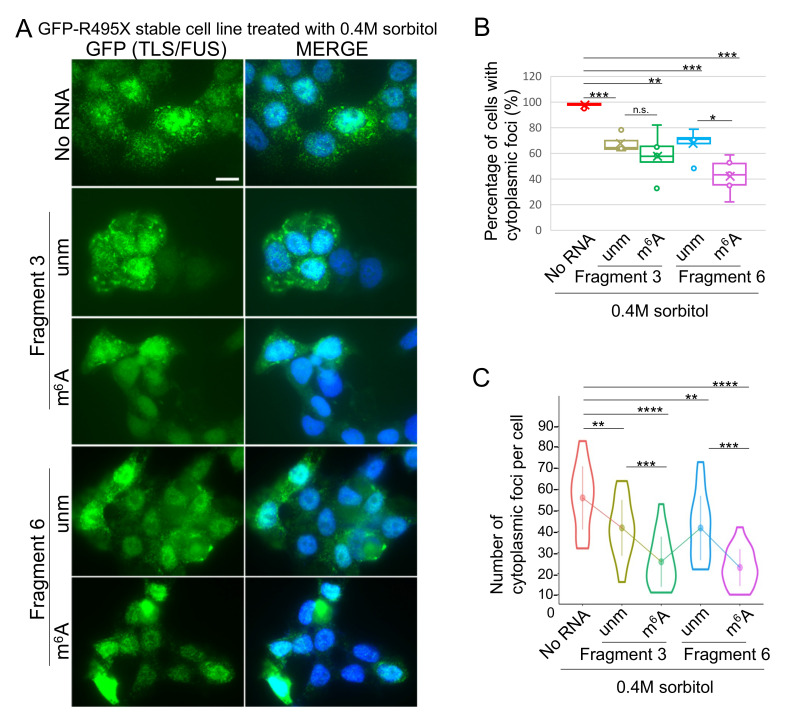
RNA fragments reduced cytoplasmic GFP-R495X foci after sorbitol treatment. (**A**) Stable cell lines expressing GFP-R495X were transfected with RNA Fragments 3 and 6 with or without m^6^A modification prior to sorbitol treatment. GFP signals were detected as TLS/FUS localization. MERGE images indicate the layered images of GFP and DAPI (for nuclei staining). Scale bar = 10 μm. (**B**,**C**) Percentage of cells with cytoplasmic TLS/FUS foci (**B**) or number of cytoplasmic foci per cell (**C**) were counted and quantified using images in (**A**) and Appendix A Appendix A. *n* = 10 (**B**) or 50 (**C**) for each sample. n.s., not significant; unm, unmodified fragments; m^6^A, m^6^A modified fragments. * *p* < 0.05, ** *p* < 0.01, *** *p* < 0.005, **** *p* < 0.001.

**Figure 5 ijms-22-11014-f005:**
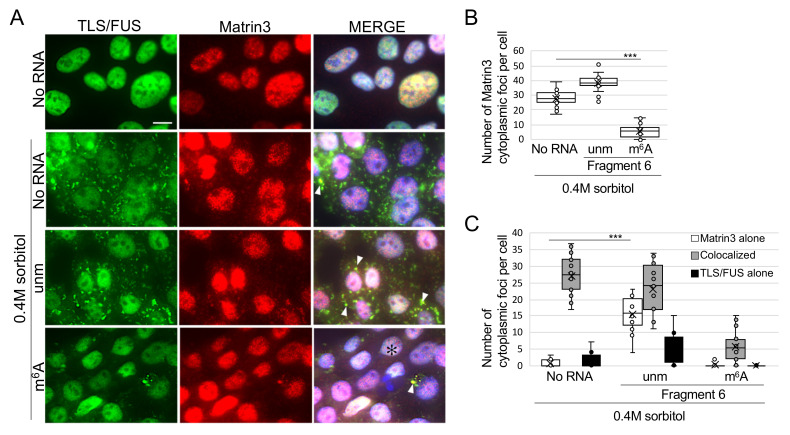
Matrin3 colocalized with cytoplasmic TLS/FUS foci after sorbitol treatment, but m^6^A-modified RNA fragment impaired this colocalization. (**A**) RNA Fragment 6 with or without m^6^A modification was transfected to HAP1 cells prior to 0.4M sorbitol treatment. Representative images of ICC are shown. MERGE images indicate the layered images of TLS/FUS, Matrin3, and DAPI (for nuclei staining). White arrowheads, colocalized TLS/FUS and Matrin3 foci; ✻, cells with cytoplasmic Matrin3 foci independent of TLS/FUS. Scale bar = 10 μm. (**B**,**C**) Cytoplasmic foci in (**A**) were quantified. *n* = 20. unm, unmodified fragment; m^6^A, m^6^A modified fragment. *** *p* < 0.005.

**Figure 6 ijms-22-11014-f006:**
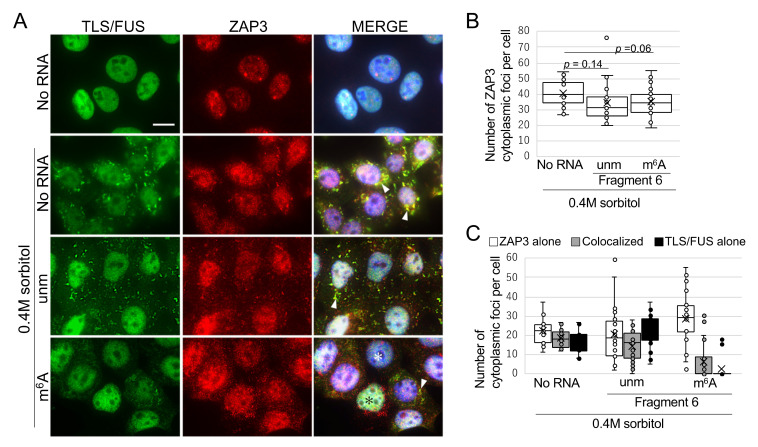
ZAP3 colocalized with cytoplasmic TLS/FUS foci after sorbitol treatment, but m^6^A-modified RNA fragment impaired this colocalization. (**A**) RNA Fragment 6 with or without m^6^A modification was transfected to HAP1 cells prior to 0.4M sorbitol treatment. Representative images of ICC are shown. MERGE images indicate the layered images of TLS/FUS, ZAP3, and DAPI (for nuclei staining). White arrowheads, colocalized TLS/FUS and ZAP3 foci; ✻, cells with cytoplasmic ZAP3 foci independent of TLS/FUS. Scale bar = 10 μm. (**B**,**C**) Cytoplasmic foci in (**A**) were quantified. *n* = 20. unm, unmodified fragment; m^6^A, m^6^A-modified fragment.

**Figure 7 ijms-22-11014-f007:**
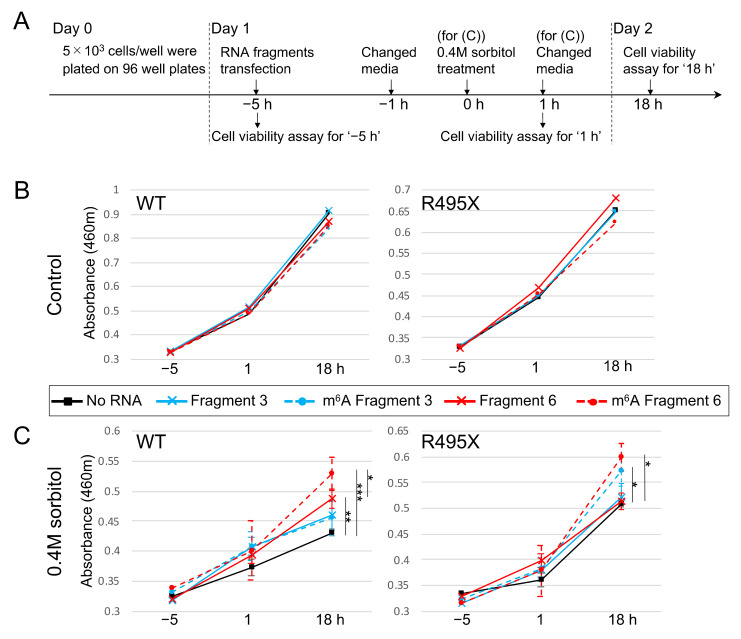
m^6^A-modified RNA fragments enhanced the viability of sorbitol-treated cells. (**A**) Experimental design of (**B**,**C**). RNA fragments were transfected prior to 0.4M sorbitol treatment, and cell viability was measured at each indicated time point. (**B**) HAP1 cells and the GFP-R495X stable cell line were transfected with indicated RNA fragments, and the cell viability was examined at each time point. (**C**) HAP1 cells and GFP-495X stable cell lines were transfected with indicated RNA fragments prior to 0.4M sorbitol treatment, and the cell viability was examined at each time point. *n* = 5. * *p* < 0.05, ** *p* < 0.01, *** *p* < 0.005.

## Data Availability

All the data are contained within the article (and the Appendix A).

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
