# Peer review of "m6A Modified Short RNA Fragments Inhibit Cytoplasmic TLS/FUS Aggregation Induced by Hyperosmotic Stress"

_ijms, 2021, doi:10.3390/ijms222011014_

Round 1

Reviewer 1 Report

The article describe the novel property of TLS/FUS protein to bind to m6A modified specific RNA fragments, and how ALS-linked mutation alters the specificity of RNA interaction both modified and unmodified by m6A methylation. Intriguingly, the author show that transfection of m6A oligonucleotide can promote survival of cells upon TLS/FUS cytoplasmic foci formation induced by sorbitol. Thus, they speculates on the possibility of using m6A modified oligonucleotide as therapeutic molecules to enhance survival one neuronal cells of ALS patients. The article is overall well written and clearly presented, pending correction of few sentences indicated in the attached PDF file. 

Nevertheless, my impression is that the impact of m6A modification on TLS/FUS  is more clear-cut and convincing for WT TLS/FUS protein then for the mutated ones. Indeed, mutant FUS/TLS isoform fails to form distinguishable foci in the cytoplasm even after sorbitol treatment. This might be a consequence of having GFP-tag fused to the protein. But WT FUS is rarely found in ALS-linked cytoplasmic granules in real ALS patients, and only the mutant forms of FUS form cytoplasmic foci and predispose to ALS. This might suggest that the proposed treatment could have little effect in real patients unless the impact of m6A oligonucleotide is proven on other ALS linked RBP.

Therefore, I would like to propose few experiments or analyses that could improve the robustness and the relevance of the study:

1) TLS/FUS aggregate have been widely shown to co-localize with stress granules constitutive factors such as TIA-1 and G3BP1. Could you please test how m6A RNA fragment alter this interaction? This could add relevance to the proposed treatment. 

2) 90% of sporadic ALS cases present cytoplasmic inclusion of the TLS/FUS homologous TDP-43 protein. Could you please test if TDP-43 interact with m6A modified RNA fragment similarly TLS/FUS by western blotting in your pool down with biotinylated RNA fragments??

3) Could you please test the impact of m6A RNA fragments on WT or mutant TLS/FUS foci formation in the absence of the GFP-tag? An indirect immunofluorescence with antibody against endogenous FUS should be performed at least for the main experiments such as the ones with sorbitol treatment. The same thing should be done in the context of survival assays: could you please test if m6A RNA fragment promote the survival of cells expressing endogenous FUS protein and not GFP tagged in a FUS KO context?

Please see additional minor comments on the attached pdf

Reviewer 2 Report

In this study, Yoneda and colleagues investigate the effect of n6-methyladenosine (m6A) on the aggregation propensity of FUS/TLS. Using 2 synthetic mRNA fragments derived from the pncRNA-D lncRNA, they show that FUS/TLS liquid-liquid phase separation (LLPS) caused by hyperosmolar stress is diminished in the presence of m6A-modified RNAs. The authors conclude that FUS/TLS is a m6A reader protein and that m6A RNAs could be potentially used to reduce FUS aggregation in patients. 
While the topic is interesting, there are several major methodological flaws that should be addressed before publication. 

  1. The authors claim that their data support the hypothesis that FUS/TLS is a novel m6A reader protein. However, that may be an overstatement since FUS interaction with m6A-modified mRNA was only tested using 2 fragments. One, Fragment 3, was not bound by WT FUS regardless of the presence of m6A modification, while the other showed an increased in binding affinity. Scrambled RNA or other mRNAs should be used to thoroughly support this statement.
  2. Most experiments lack important controls. Figures 2, 5, 6, and 7 test the effect of the modified or unmodified mRNA on the behavior of WT or mutant TLS. However, the authors compare endogenous WT FUS to cells stably expressing GFP-tagged mutant FUS. That is not appropriate, as the presence of the GFP-tag and the different levels of expression between endogenous and transgenic FUS/TLS may have a large influence on FUS localization and binding patterns.  
  3. A “no RNA” or “scramble RNA” control should be included for most experiments.
  4. It is unclear why cy5-labeled RNA has not been used in all experiment to assess FUS and mRNA colocalization.
  5. Figure 4B shows that Fragment 6 has the biggest effect on FUS foci formation. However, the images shown in panel A show Fragment 3 being more effective.
  6. Only one stressor (hyperosmolar stress) was tested to induce FUS LLPS. It is uncertain whether similar effects could be seen under different experimental conditions, and what the significance of these finding may be to human disease
  7. The authors state that ALS-linked FUS mutants have reduced binding to m6A modified mRNA in Figure 1. However, the data presented do not support that conclusion. Statistical analysis to cross compare WT and mutants would be required.
  8. Figure 5 and 6, in which FUS colocalization with Matrin 3 and ZAP3 is shown, should be quantified.
  9. Statistical analysis of the data should be revised, as multiple comparisons must be tested via one-way ANOVAs and not t-tests.
  10. Effect of sorbitol treatment on FUS R521G should be shown
  11. In section 4.2, line 579, ref# should be 60, not 30

Reviewer 3 Report

Yoneda et al. investigated that m6A recognition protein can modified RNA fragments to enhance cell viability. Overall, the research work is excellent. Thus, I only have mild suggestions for this study.

  1. The schematic drawing of the region around cyclin D1 promoter in Figure 1A is modified by the researcher’s previous publication (Figure 4B in J Biol Chem. 2020 Apr 24;295(17):5626-5639). Modification is recommended to avoid duplication.

  1. Some grammar errors

[1] Page 2, Line 88, “a“ similar affinity to both Fragment 3 and 6

[2] Page 2, Line 91, plays “an” important role in RNA recognition

[3] Page 3, Line 145, formed “a” small number

[4] Page 6, Line 268, data clearly “correlated” with the binding intensity

[5] Page 7, Line 324, smaller “numbers” of cytoplasmic foci

Round 2

Reviewer 1 Report

The authors have added the additional staining and controls and made figures more clear.

Nevertheless major experiments have not been performed to address if GFP tag can alter the affinity to m6A modified RNA fragments. This doubt still persist

The major negative control is No RNA as better indicated in the figures, nevertheless the usage of an unrelated sequence +/- m6A modification would strengthen the conclusion if the interaction of FUS protein with the modified RNA is specific for the fragment's sequence or not

In conclusion I have the impression that the manuscript has been improved marginally but that the study is still relevant fir the scientific comunity

Reviewer 2 Report

In this revision, the authors have done minimal work to address the concerns raised in the previous review. Though editing of the results and conclusions, the authors have resolved some of the major inconsistencies, but have not address the core issues such as lack of control. By toning down their conclusion to better fit the results, they have reduced the significance and potential impact of the paper, which could have been improved and expanded with a few additional experiments. 
